# Enhanced validation of antibodies for research applications

Fredrik Edfors [1,2], Andreas Hober [1,2], Klas Linderbäck[2], Gianluca Maddalo[1,2], Alireza Azimi[3], Åsa Sivertsson[1,2], Hanna Tegel [2], Sophia Hober[2], Cristina Al-Khalili Szigyarto[1,2], Linn Fagerberg [1,2], Kalle von Feilitzen[1,2], Per Oksvold [1,2], Cecilia Lindskog[4], Björn Forsström [1,2] & Mathias Uhlen [1,2,5]

There is a need for standardized validation methods for antibody specificity and selectivity. Recently, five alternative validation pillars were proposed to explore the specificity of research antibodies using methods with no need for prior knowledge about the protein target. Here, we show that these principles can be used in a streamlined manner for enhanced validation of research antibodies in Western blot applications. More than 6,000 antibodies were validated with at least one of these strategies involving orthogonal methods, genetic knockdown, recombinant expression, independent antibodies, and capture mass spectrometry analysis. The results show a path forward for efforts to validate antibodies in an application-specific manner suitable for both providers and users.

[1] Science for Life Laboratory, KTH - Royal Institute of Technology, SE-171 21 Stockholm, Sweden. [2] Department of Protein Science, KTH - Royal Institute of Technology, SE-106 91 Stockholm, Sweden. [3] Department of Oncology-Pathology, Karolinska Institute, Karolinska University Hospital, Stockholm SE-171 77, Sweden. [4] Department of Immunology, Genetics and Pathology, Uppsala University, Uppsala SE-751 85, Sweden. [5] Novo Nordisk Foundation Center for Biosustainability, Technical University of Denmark, DK-2970 Hørsholm, Denmark. Correspondence and requests for materials should be addressed to M.U. (email: mathias.uhlen@scilifelab.se)

There are several challenges that must be overcome to validate the specificity and reproducibility of antibody reagents and this has been emphasized by many recent publications[1–8]. One important issue is cross-reactivity due to off-target binding, which is defined as antibodies binding to proteins other than the intended target[9]. Another issue is that samples are treated differently in different applications, which influence the epitopes exposed on the target protein. As a result, this might have profound consequences for the ability of a given antibody to bind specifically to its target[10]. Thus, antibodies must be validated in an application-specific manner, as was recently pointed out by the International Working Group for Antibody Validation (IWGAV)[11]. The working group proposed five pillars for antibody validation, all allowing the antibody to be validated without the need for any prior knowledge of the target protein, except the gene and protein sequence. This is important since many of the proteins predicted from the genome sequence lack previous literature and they are thus important targets for antibody-based studies to allow a genome-wide analysis of the entire human proteome. Here, we have adapted these pillars for Western blot applications with a focus on providing scalable and streamlined methods. We present a strategy for enhanced validation suitable for commercial providers as well as small sized research groups with limited resources.

The Western blot application is the most frequently used antibody-based method[12], with approximately 1.5 million antibodies classified as supported for this application in the Antibodypedia portal[13]. In a Western blot assay, the approximate size of the target protein is obtained as part of the analysis and off-target binding can be probed by the presence or absence of additional bands. However, many protein bands are shifted in size compared to the predicted molecular weight, e.g. due to proteolytic processing and various post-translational modifications, including glycosylation. Yang et al.[14] cataloged such variation by carrying out proteome-wide quantitative mass spectrometry (MS) in a vast number of slices from polyacrylamide gels and reconstructing virtual Western blots that were compared to predicted molecular weights of unmodified proteins. Around 15% of the proteins had their most prominent band far from the predicted molecular weight and even more had multiple bands that all mapped to the same protein-coding gene. Consequently, there is an obvious need for validation principles that are independent and complementary to the theoretical size estimate used in the standard Western blot assay.

Here, we show that the methods described can be used for streamlined validation of antibodies for Western blot applications using convenient panels of cell lines for the analysis. More than 6,000 antibodies were validated in at least one of the five methods and all the primary data for the validation is presented as part of the Human Protein Atlas (www.proteinatlas.org) antibody info page for each respective antibody A special focus was to investigate the performance of the orthogonal and gel-migration capture MS validation strategies, since there are no prior examples in the literature of systematic validation of antibodies using these two rationales. In the orthogonal validation strategy, the protein abundance levels obtained using an antibody-dependent method are compared with the levels determined by an antibody-independent method across a set of samples. For the capture MS strategy, the apparent size obtained by an antibody is compared to the presence of MS-determined target peptides after cutting out gel slices. Altogether 1,630 antibodies were validated by at least two of the pillars and 267 were validated by three or more pillars. The results show a path forward for streamlined validation of antibodies, although it is important to point out that the enhanced validation is specific for a certain sample context and the validation is thus dependent on the sample preparation procedures used to evaluate the assay including the relative abundance of the target protein.

## Results

**Orthogonal validation of antibodies using proteomics.** More than 6,000 antibodies were validated in at least one of the five methods and all the primary data for the validation is presented as part of the Human Protein Atlas (www.proteinatlas.org) antibody info page for each respective antibody (Supplementary Data 1). In Fig. 1, a description of the five validation methods adapted for validation of research antibodies in Western blot applications are outlined including a summary of the features and limitations of each strategy. For orthogonal validation, the protein levels determined by an antibody-dependent method are compared with the levels determined by an antibody-independent method across a panel of samples. Here, we decided to use a panel of cell lines (Fig. 2a) to allow for protein expression analysis with the aim to establish a platform for validation that could be easily reproduced by both users and providers of antibodies. We decided to select cell lines (Supplementary Table 1) from a list of 56 cell lines (Supplementary Fig. 1) showing highly variable gene expression levels based on genome-wide transcriptomics analysis

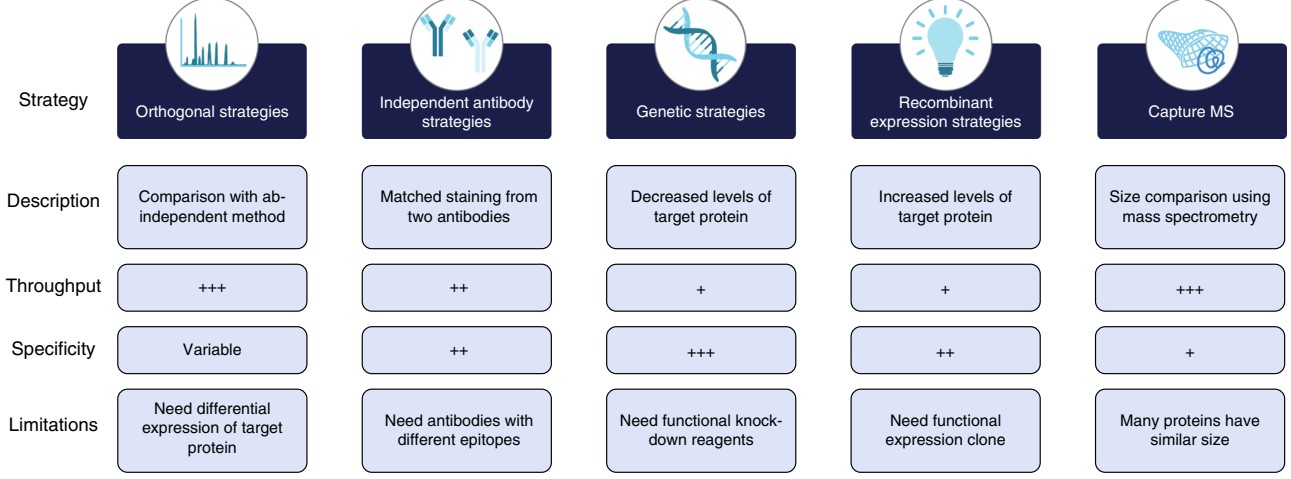

**Fig. 1** Advantages and disadvantages of the five strategies used for enhanced Western blot antibody validation. Throughput is a relative estimate of the possibility to use the method for high-throughput settings and specificity is the estimate of how reliable the validation results are to determine the specificity of the antibody. Low: +, Medium: ++, High: +++

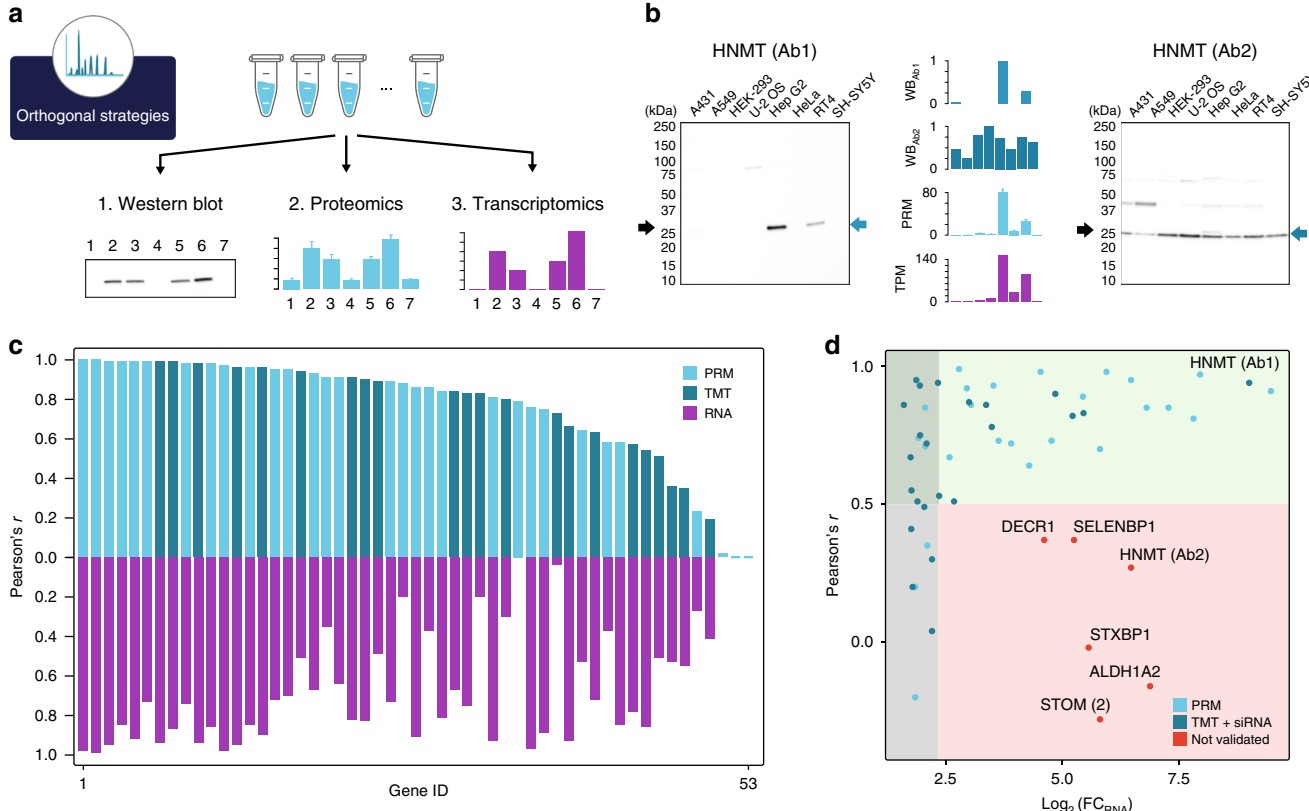

**Fig. 2** Orthogonal validation of antibodies using proteomics. **a** Principle for the Western blot validation strategy based on correlating protein expression levels determined across a panel of cell lines using either proteomics or transcriptomics. **b** Example of orthogonal validation of Western blot bands (WB, relative intensity) by proteomics (Parallel Reaction Monitoring, PRM) reported as ratio to standard or transcriptomics reported as Transcript Per Million (TPM). Error bars represent 1 s.d. across three technical replicates. The black arrow indicates the theoretical molecular weight of the protein and blue arrows indicate the band subjected for the intensity-based relative quantification to determine the antibody staining profile. More examples including loading controls are presented in Supplementary Fig. 2–3 for PRM and TMT, respectively. **c** Mirror plot summarizing the Pearson's r for 53 antibodies evaluated either by TMT (dark blue) or by PRM (light blue) including transcriptomics-based validation based on RNA expression (purple). **d** Analysis of the Pearson's r between Western blot band intensities and RNA expression levels as a consequence of the fold-change between the highest and lowest value across the cell lines. The gray area represents fold-change in RNA levels less than fivefold. Antibodies in the green area (Pearson's r > 0.5) are considered validated while antibodies in the red area (Pearson's r < 0.5) are considered not validated

(Supplementary Data 2). The analysis of the human cell lines demonstrates that a majority ($n = 14,622$) of all the protein-coding genes ($n = 19,628$) have a relatively high RNA expression (TPM value > 10) in at least one of the cell lines.

We first explored the possibility to validate antibodies for Western blot application by orthogonal validation using a cell line panel quantitatively measured by MS-based proteomics, either by an unbiased tandem mass tag (TMT) 10plex shotgun strategy[15] or by a target proteomics Parallel Reaction Monitoring (PRM) strategy (Supplementary Data 3-4). The Western blot bands for each antibody could thus be evaluated by correlating the band intensities with the proteomics results (Supplementary Data 5). For the TMT-based proteomics analysis, 23 antibodies towards 22 protein targets were analyzed, including seven membrane-bound proteins, while 33 antibodies towards 29 protein targets were analyzed and quantified using a targeted proteomics approach (PRM) with internal standard spike-in. In Fig. 2b, the results using two antibodies towards a histamine N-methyltransferase (*HNMT*) are shown for the targeted proteomics approach (PRM) demonstrating high correlation with the levels determined by Western blot band analysis for antibody 1, while the second antibody shows one band of similar size, but no correlation with the proteomics results across the cell lines. Thus, antibody 1 is validated for Western blot analysis, unlike antibody 2, which is

not validated for this application. In Supplementary Table 2, a summary of the results for all antibodies are shown and the correlation results (Supplementary Table 3) are visualized in Fig. 2c (blue bars). If a Pearson correlation cut-off of 0.5 is considered across the cell panels, 46 of the antibodies pass the validation criteria, while 7 antibodies have a correlation below 0.5.

**Orthogonal validation of antibodies using transcriptomics.** An alternative to MS-based proteomics is to use transcriptomics analysis as an orthogonal method (Fig. 2a). This approach is based on the hypothesis that mRNA levels and corresponding protein levels correlate at the steady state, which has been suggested by several publications[16–19]. This method paves the way for a relatively simple validation scheme for antibodies, since transcriptome analysis of a set of reference cells and tissues provides a data set that can be used in a proteome-wide manner. Here, we generated transcriptomics data for all the cell lines used in the validations described above. This enabled us to also use the transcriptomics data for validation of the antibodies analyzed above as shown for the examples in Fig. 2b. It is reassuring that the validation of the two antibodies towards *HNMT* follows the proteomics results with antibody 1 validated, in contrast to antibody 2. A summary of the transcriptomics validation for all 53 antibodies is shown in Fig. 2c (purple bars) with 39 of these

antibodies passing using a Pearson correlation higher than 0.5 across all cell lines as cut-off.

To explore the correlation-based evaluation further, a scatter plot (Fig. 2d) was generated displaying the dependence of correlation as a result of RNA-level difference ($log_2$ fold-change between highest and lowest expression level). The same type of plot showing the correlation as a function of difference in protein levels for PRM and TMT can be found in Supplementary Fig. 4. The plot (Fig. 2d) shows that six antibodies do not pass the evaluation based on a cut-off of 0.5 despite a higher than fivefold difference in RNA levels across the cell lines. These were thus scored as not validated for Western blot applications. The fold-change in the RNA levels was less than fivefold for eight antibodies, resulting in a low correlation coefficient, <0.5, despite that they previously had been validated by proteomics. This suggests that the RNA-based method requires relatively higher variability of expression levels to achieve the required correlation coefficient. To further validate these antibodies, we performed a genetic knockdown using gene-specific siRNA reagents in the cell line U-2 OS for all antibodies evaluated using the TMT-based proteomics analysis, including the set of 14 antibodies with less than fivefold change expression (Supplementary Fig. 2). The results confirm that the antibodies which had a low variability and failed the transcriptomics-based method are indeed specific for the protein target supporting the positive validation by the proteomics methods.

In summary, the orthogonal methods allowed us to validate 46 of the 53 antibodies as specific for the target protein in the Western blot assay, although six of the antibodies could not be validated by the transcriptomics method, since the low expression variability for these target proteins resulted in high statistical noise and thus low formal correlation. The results demonstrate the dependence on expression variability for correlation studies.

**Orthogonal validation using a standardized assay**. We next decided to perform a systematic orthogonal validation based on proteomics for antibodies used in the Human Protein Atlas (www.proteinatlas.org) program, in which a standard assay using two cell lines (RT4 and U-251) has been carried out. Since many Western blots contain several bands, we decided to use the criteria to only validate the strongest stained band in each case and this band needed to be clearly separated from the weaker bands (if present). In addition, as suggested above, we required a fivefold difference in RNA levels between the two cell lines for unambiguously allowing validation by the orthogonal method, but we also allowed the antibody to be correlated if the expression levels (trend) in the two cell lines supported the correlation and the size of the band was confirmed either by theoretical size estimate or by the capture MS method described above. In Supplementary Fig. 5 an example of an antibody towards *AASS* is shown supported by both the orthogonal methods (proteomics and transcriptomics). Similarly, the stronger band for the antibody towards *SLC9A3R2* is validated using both orthogonal methods, although a weaker band of higher molecular weight is present in both cell lines. Using these criteria, 711 antibodies were validated using the TMT proteomics data set (Supplementary Data 6).

**Capture MS validation**. An alternative proteomics-based method for antibody validation is to use a virtual Western protocol as outlined by Yang et al.[14]. This method is based on comparison between the molecular weight of the stained band visualized by the antibody against the protein size obtained by a capture MS method in which multiple gel slices are cut out from the electrophoretic separation and analysed separately by proteomics (Fig. 3a). We decided to generate these virtual Western blots

using cell extracts from the two cell lines U-251 and RT4 used in the standardized Western blot validation in the Human Protein Atlas effort. Protein extracts for these cell lines were individually separated by SDS-PAGE in three technical replicates and 50 slices were excised for each lane, proteolytically digested by trypsin and analyzed by shotgun proteomics. The peptides of the target protein could thus be identified in the different gel slices and general virtual Western blots could be generated covering a large part of the human proteome. In total, we identified proteins towards 5,523 antibodies (Supplementary Data 7) in at least one of the two cell lines allowing these antibodies to be validated by the capture MS method. In Fig. 3b, an example of such a comparison is shown with the size distribution of the protein-specific peptides determined by the capture MS method shown and a band of the expected size is confirmed in both cell lines based on gel migration.

Altogether, 3,650 antibodies generated within the Human Protein Atlas effort were subsequently analyzed using the orthogonal data from the capture MS approach and a summary of the results are shown in Fig. 3c. In total, 2,888 (79%) of the analyzed antibodies were shown to be supportive using the capture MS approach (Supplementary Data 6). For approximately half of these antibodies ($n = 1,517$), the antibody was also scored as enhanced using at least one additional validation strategy.

The size estimates determined in both assays are relatively crude and many proteins have similar molecular weights and we therefore decided to analyze the theoretical molecular weights of the entire human proteome as predicted from the genome sequence (Fig. 3d, Supplementary Data 8). The results show that for the human proteome, there is a maximum of proteins with sizes around 50 kDa, corresponding to a length of ~450 amino acids.

**Independent antibody for validation**. The independent antibody strategy for antibody validation is based on comparison of the staining pattern using two independent antibodies with no overlapping epitopes. The same criteria as outlined above for orthogonal validation was used and we have used the whole panel of five samples from the standardized Human Protein Atlas Western blot set-up with two cell lines (RT4 and U-251), one sample of pooled human plasma and two tissue extracts (liver and tonsil). Several examples of independent antibody validation are shown in Supplementary Fig. 6 and altogether 1,922 antibodies were validated using this method (Supplementary Data 6).

**Genetic strategies for antibody validation**. We subsequently decided to use the genetic method for validation. The staining of the antibody is evaluated by Western blot through analyses of samples from cell lysates before and after knockdown of the corresponding target gene. Here, we have compared the staining of samples before and after knockdown of the target protein in the human cell line U-2 OS by using two siRNA reagents specific for the target gene. At least one of the reagents must result in a down-regulation of more than 25% of the target protein. Several examples are shown in Supplementary Fig. 7 and altogether 217 antibodies were validated using this method (Supplementary Data 6).

**Recombinant expression for validation of antibodies**. Finally, we used a recombinant expression strategy to over-express the target proteins that in most cases were not expressed in the cell lines used in the standard pipe-line. The staining of the antibody is evaluated by Western blot through analyses of samples from cell lysates with and without recombinant expression of the target protein. Here, we have used recombinant expression in the

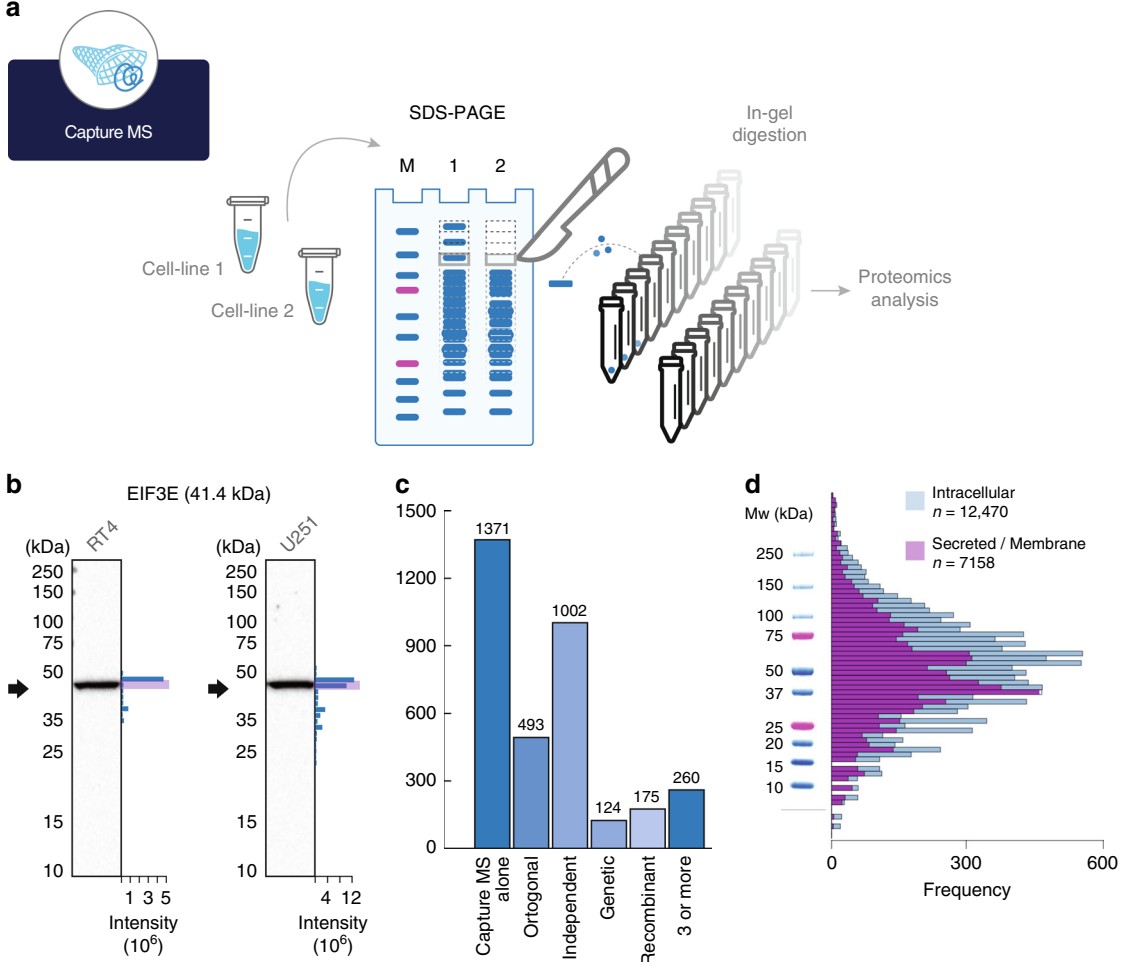

**Fig. 3** Validation of antibodies using capture MS. **a** Principle for validation of antibodies based on separation by SDS-PAGE and comparing the protein migration profile determined by an antibody (Western blot) and peptide identification performed by proteomics (capture MS). **b** Examples of orthogonal validation of two antibodies by capture MS in two different cell lines (RT4, U-251). The black arrow indicates theoretical molecular weight, and the blue bars represent number of peptides identified in each gel slice. **c** Summary of capture MS validation of antibodies ($n = 2,888$) showing if they were validated using other antibody validation pillars. **d** Distribution of theoretical molecular weights for the largest transcript of all ($n = 19,628$) human protein-coding genes divided into intracellular (blue) and secreted and membrane bound proteins (purple). Molecular weights are represented by the $\log_2$ value together with a molecular weight ladder from a typical Western blot assay used within the Human Protein Atlas program

human cell line HEK 293 and samples with and without expression vectors have been analysed by Western blot. The antibody must show a strong band in the cell line with recombinant expression and no or fainter band in the control. Several examples of this validation method are shown in Supplementary Fig. 8 and altogether 2,190 antibody targets were validated using this method (Supplementary Data 6).

**Summary of the enhanced validation of antibodies**. The results of the enhanced validation using the five strategies (Supplementary Table 4) are shown in Fig. 4a and summarized in Supplementary Table 5 and individually listed in Supplementary Data 6. Altogether 6,014 antibodies have been validated by at least one of the enhanced validation strategies. Note that only orthogonal validation using proteomics have been included in this summary. A summary of 263 antibodies validated by three or more pillars is shown in Fig. 4b and the validation in each pillar has been scored as enhanced, uncertain or not done. The category uncertain indicates that the particular validation strategy has been tried, but the result of the validation is inconclusive, in most cases due to the nature of the target protein. Many antibodies that show low

correlation in the orthogonal assay due to low variability of the target protein and many antibodies in the capture MS assay fall just outside boundaries of the expected band, which could be a result of uncertainty in cutting out bands on a gel. Some antibodies were scored as uncertain for the independent antibodies validation and the lack of validation was in all cases due to the non-specific staining of the paired antibody. For the genetic method, some of the antibodies were scored uncertain and although not proven, the lack of disappearance of a band in the assay is most likely caused by failure of the siRNA reagents to knockdown the target protein and thus due to technical issues related to the genetic method, not the antibody specificity. Finally, for the recombinant expression validation, only two antibodies were scored as uncertain and, in both cases, this was due to the lack of expression in the recombinant cell line.

## Discussion

We show a path forward for systematic validation of antibodies based on the five pillars proposed earlier[11], but here adapted for Western blot applications. The fact that each of the validation pillars have limitations (Fig. 1) enforces the need for careful

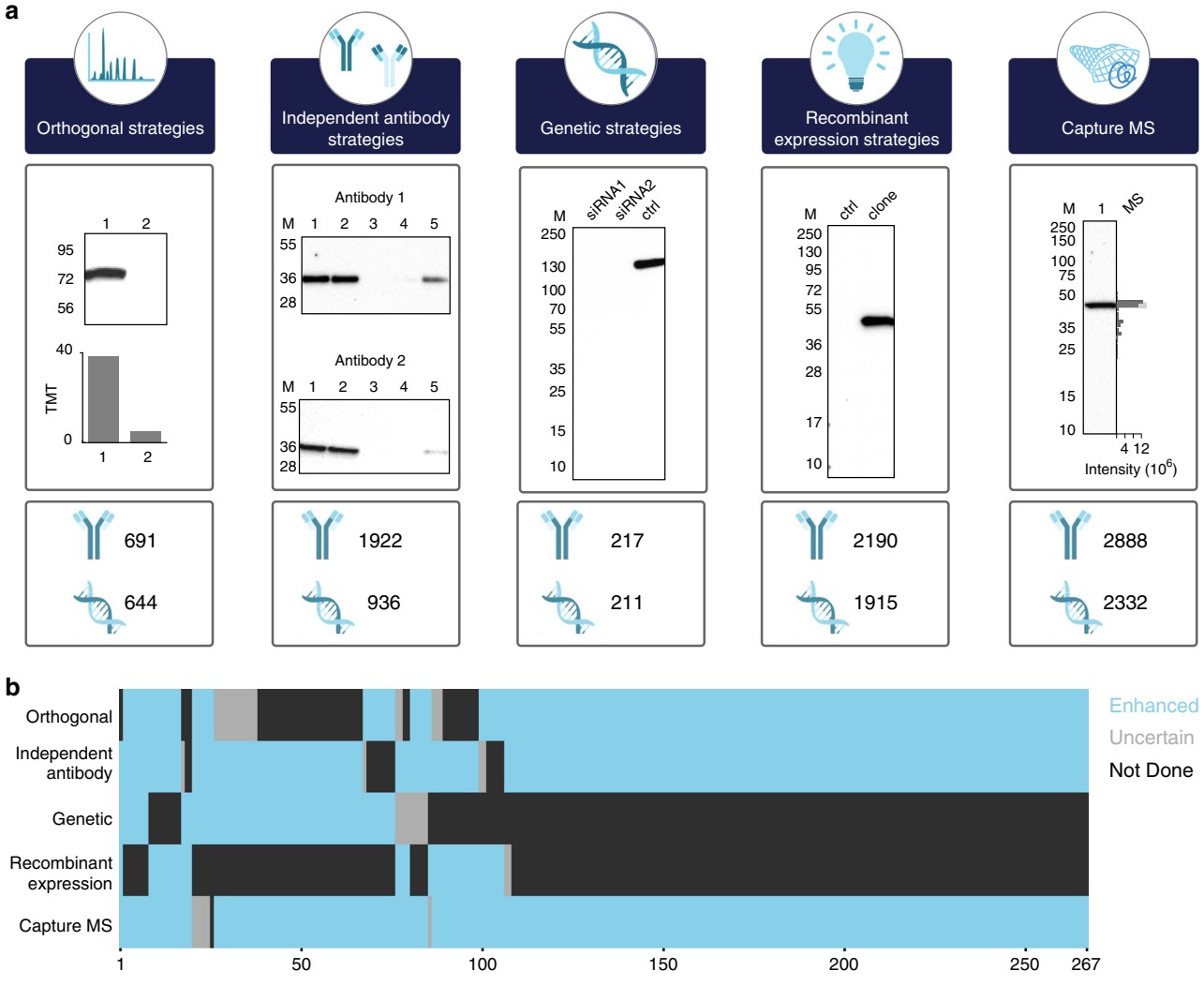

**Fig. 4** Summary of the five-pillar strategy used for antibody validation. **a** Summary of the five-pillar strategy employed within the Human Protein Atlas with the number of enhanced validated antibodies together with gene-coverage in each respective category. **b** Heat map showing 267 antibodies with enhanced validation (blue) in more than three of the five pillars. The two other validation states include not done (black) and uncertain (gray)

assessment of the most suitable strategy for a particular antibody. Hence, the capture MS and the orthogonal methods are highly suitable for streamlined efforts, since the underlying analysis using mass spectrometry or transcriptomics can be performed in advance in a centralized manner and the supporting data can be shared through an open access policy. However, both methods have limitation regarding specificity for some of the target proteins in certain sample contexts. In contrast, independent antibodies do not need any special reagents, clones or data from additional methods, but since this method relies on knowledge of the binding epitope of respective antibody, it cannot be used for most antibodies available today. Genetic approaches based on both CRISPR and siRNA are attractive and yield, in most cases, easily interpreted results, but many proteins are essential for the cell and these can therefore not easily be knocked down and most validation efforts based on a genetic method only involves one single cell line, which of course limits the broadness of the validation. It is also difficult to use genetic methods for validation of antibodies in human tissues and organs, limiting this validation strategy for immunohistochemistry applications. Thus, many antibodies can only be validated by a restricted number of the five validation principles due to the inherited nature of the target protein, i.e., similar expression level (excludes orthogonal

validation), no detectable peptides in proteomics (excludes capture MS), lack of expression in the recombinant host (excludes recombinant expression), or that the target protein is essential for the cell (excludes genetic). In summary, antibodies towards housekeeping proteins expressed at similar levels in all cells, are well suited for validation using genetic and/or capture MS strategies, while target proteins showing high variability in at least two cells used for the validation, are well suited for orthogonal and/or independent antibody strategy. In contrast, if the target protein is not expressed at all in the panel of cells used for the validation, it might be preferable to use the recombinant expression strategy.

It is important to point out that although each of the five enhanced validation principles provide evidence for specific protein target binding by the antibody with no or little off-target binding, this should not be understood as that the antibody will be specific in all sample contexts. The wording good and bad antibody or the most specific antibody should be avoided, since a specific antibody in one sample context can give rise to high cross-reactivity in another sample context depending on the nature of the epitope(s) that it will recognize. The wording enhanced should thus be interpreted that the antibody has been validated in a certain sample context by at

least one of the pillars and it is therefore important that the primary data describing the validation is provided in an open access manner.

Here, many of the antibodies were classified as uncertain and in most cases this was probably due to limitations of the validation assays for the respective target protein rather than the poor performance of the antibody. This demonstrates the value of using different independent validation methods without the need of any prior knowledge of the target protein, and the data suggest that the validation in one pillar yields enough comfort that the antibody is specific for the target, although the addition of even more validation pillars supports the specificity of the antibody even further. We show that more than 6,000 antibodies published on the Human Protein Atlas could be validated in this enhanced manner using at least one of the five strategies presented here and more than 1,600 of these antibodies were validated by at least two of the pillars and 267 with three or more pillars (Fig. 4b). It is important to point out that the staining of bands in Western blot assays cannot be considered as strict quantitative. This means that strict criteria for the stringency to consider an antibody validated is not easy to define. It is therefore important to show the primary data used for the validation enables users of a particular antibody to review the supporting data. Here, all the underlying primary data to support the enhanced validation score are published as part of the antibody information page for respective antibody in the Human Protein Atlas. In addition, we provide the capture MS data for the two cell lines used here (RT4 and U-251). This includes identification and label-free quantification of proteins and their migration in the gel for 5,605 human target proteins. This data can now be used by providers and users to validate antibodies if these cell lines are used. Similarly, the transcriptomics data for all cell lines used in any of the panels in this work are publicly available in an open access manner to allow providers and users to perform orthogonal validation of their antibodies using these cell lines.

A convenient strategy for validation of antibodies is to use a capture MS strategy based on identifying target protein peptides in digested sections of the electrophoresis separation gel by mass spectrometry and to compare the proteins' migration to the Western blot bands. An advantage of this method is that the proteomics data only need to be generated once for a particular cell line and the proteomics data for thousands of target protein peptides for a selected cell line can be readily shared among research groups in an open access manner (Supplementary Data 9). This enables the use of data from a single shotgun proteomics experiment to validate thousands of antibodies in different laboratories. An additional strength of the method is that it can prevent antibodies from being wrongfully rejected by confirming their migration pattern when this differs from the target proteins' theoretical molecular weight. However, we consider the capture MS validation the weakest of the validation pillars described here, due to the fact that many proteins are migrating to the same region of the Western blot gel. The capture MS validation is even more unreliable for mid-sized proteins of molecular weight between 30 and 70 kDa emphasizing the need for extra caution when analyzing Western blot bands in this size interval, due to the added risk of off-target binding to unrelated proteins of similar sizes.

One of the focuses in this work has been to introduce orthogonal methods for validation of antibodies. This strategy opens up streamlined efforts to validate antibodies using verified and standardized cell or tissue lysate panels centrally obtained by certified providers, in which the antibody-independent method is performed and the data used for validation is provided and shared in an open access manner. One of the issues of abundance correlation is that cell lines with variable expression of the target

protein must be selected in the analysis. However, genome-wide transcriptomics data is now available for thousands of cell lines and this facilitates the selection of suitable cell lines for a particular protein target. Here, we show that the TMT proteomics approach yields protein abundance data for 6,733 protein targets using only two cell lines and similarly, the transcriptomics yields essentially whole-genome expression data in a quantitative manner for more than 14,000 protein-coding genes, thus enabling validation of a large fraction of the protein-coding genome using just a few selected cells or tissues. It is reassuring that all the antibodies validated by the transcriptomics method were subsequently also validated by the orthogonal proteomics approaches and/or the siRNA knockdown, suggesting that this method did not generate false positives and thus makes it attractive for streamlined validation efforts. A convenient systematic scheme can thus be designed, suitable for analysis of a large fraction of the human protein targets.

We show that all the validation strategies described here can be used to investigate and validate antibodies that yield several bands in the Western blot assay. This phenomenon is not uncommon and could either be due to unspecific staining as a result of off-target binding or specific staining of multiple isoforms of the target protein due to proteolysis, post-translational modifications or splice variants. The question arises if such an antibody should be approved for Western blot application, provided that the off-target band can be readily separated from the band corresponding to the target protein. An important issue is thus how to annotate and score these antibodies that have been validated by a two-dimensional analysis combining size on the Western blot with one of the validation strategies presented. Here, we decided to allow the enhanced validation of antibodies with several bands present, but only if the validated band is the most stained and that it is clearly separated from any weaker band(s).

In summary, we show that the five pillars outlined in Fig. 1 can be used to validate antibodies for Western blot applications in a standardized and systematic manner. The methods described here can also be used for other antibody-based methods, such as immunohistochemistry, immunocytochemistry, flow sorting, and sandwich assays. Note that the validation must be performed in an application-specific manner to ensure that the antibody validation is relevant for the sample treatment of each application. In this context, it is important to point out that the use of a panel of cell lines or tissues in the validation effort might not detect off-target binding in other sample contexts. In many cases it might be preferable to use a target-specific panel of tissues or cells to ensure variability across the samples and thus making correlation studies more accurate.

## Methods

**Production and quantification of protein standards**. DNA fragments were initially cloned into the expression pAff8c[20] and thereafter transformed into an *E. coli* strain, auxotrophic for the amino acids arginine and lysine[21], for recombinant protein production. Cells containing expression vectors were cultivated in 10 ml minimal media using 100 ml shake flasks[20]. Heavy isotope-labeled ($^{13}C$ and $^{15}N$) versions of lysine and arginine (Cambridge Isotope Laboratories, Tewksbury, MA, USA) were provided to the cells at 200 μg/ml to generate fully incorporated heavy protein standards. Cell cultures were harvested and the QPrESTs were purified using the N-terminal quantification tag (Q-Tag), which included a hexahistidine tag used for purification by Immobilize Metal Affinity Chromatography (IMAC). After purification, all isotopic QPrEST fragments were absolutely quantified by mass spectrometry against a non-labeled purified Q-Tag standard. The Q-Tag standard, also including a C-terminal OneStrep-tag. This protein was purified using IMAC chromatography and the IMAC elution buffer was exchanged for 1 × PBS (10 mM NaP, 150 mM NaCl, pH 7.3) using a PD-10 desalting column (GE Healthcare, Uppsala, Sweden). The sample was thereafter purified on a StrepTrap HP column (GE Healthcare) on an ÄKTA Explorer system (GE Healthcare) according to the manufactures protocol. All QPrEST fragments were quantified by mixing 1:1 with Q-Tag-standard, which previously had been quantified by amino acid analysis. The protein mixture was digested by trypsin by

first reducing the proteins with 10 mM dithiothreitol (DTT) for 30 min at 56 °C. This was followed by addition of 50 mM iodoacetamide (IAA) and incubated in dark for 20 min. Proteomics grade porcine trypsin (Sigma-Aldrich, St Louis, MO, USA) was added in a 1:50 enzyme to substrate ratio and incubated in a thermo-mixer at 37 °C. The reaction was quenched after 16 h by addition of formic acid (FA) and the sample was desalted using in-house prepared StageTips packed with C18 Bonded Silica matrix (3 M Empore)[22]. Briefly, three layers of Octadecyl membrane were placed in 200 μl pipette tips. The membrane was activated by addition of 100% acetonitrile (ACN), followed by centrifugation for 1 minute at 961 rcf. The membrane was equilibrated by addition of 0.1% FA, Milli-Q (MQ) followed by centrifugation for 1 min at 931 rcf. The sample was acidified prior to addition onto the membrane, followed by centrifugation for 1 min at 931 rcf. The membrane was washed twice with 0.1% FA, MQ and the peptides were eluted in two steps using 60% ACN, MQ. Desalted peptides were vacuum dried before subjected for LC-MS/MS analysis.

**Cell line cultivation used for targeted proteomics**. Eight different cell lines (A431, Hep G2, A549, HeLa, HEK 293, RT4, SH-SY5Y [available from DSMZ] and U-2 OS [LGC/ATCC],) were cultivated at 37 °C in a humidified atmosphere containing 5% $CO_2$. A549 and SH-SY5Y were cultivated in Dulbecco's modified Eagle's Medium (Sigma-Aldrich). HEK 293, HeLa and Hep G2 were cultivated in Minimum Essential Medium Eagle (Sigma-Aldrich). U-2 OS and RT4 were culti-vated in McCoy's medium (Sigma-Aldrich) and A431 was cultivated in RPMI-1640 (Sigma-Aldrich). All media were supplemented with 10% Fetal Bovine Serum (Sigma-Aldrich). Media for HEK 293, HeLa and Hep G2 were supplemented with 1% MEM non-essential Amino Acid Solution (Sigma-Aldrich) and media for and Hep G2 were also supplemented with 1% L-Glutamine (Sigma-Aldrich). The cells were cultivated up to 80% confluence and cell numbers were counted with a Scepter 2.0 Cell Counter (Merck Millipore, Billerica, MA, USA) before pellets were collected and stored at −80 °C.

**Cell lysis for targeted proteomics**. Cells were dissolved in Lysis buffer (100 mM Tris-HCl, 4% SDS, 10 mM DTT, pH 7.6) and incubated at 95 °C in a thermomixer for 5 min at 600 rpm and thereafter sonicated at 50% amp (1 s pulse, 1 s hold) for 1 min. The same cell lysate was subjected for mass spectrometry analysis and Western blot analysis.

**Filter-aided sample preparation**. One QPrEST mastermix was prepared to represent endogenous levels of U-2 OS with light to heavy peptide ratios as close to one as possible. The same amount of mastermix was spiked into cell lysates con-sisting of one million cells. The lysate was diluted with denaturing buffer (8 M Urea, 100 mM Tris-HCl, pH 8.5) and centrifuged through a 0.22 μm spin filter (Corning, Corning, NY, USA). Trypsin digestion was performed using a previously described filter-aided sample preparation (FASP) method[23]. Trypsin was added in a 1:50 enzyme to substrate ratio and incubated overnight at 37 °C. Peptides were extracted by solid-phase extraction using in-house prepared C18 StageTips following the protocol described above. Desalted peptides were vacuum dried before LC-MS analysis.

**Liquid chromatography for targeted proteomics**. Liquid chromatography was performed using as Ultimate 3000 binary RS nano system (Thermo Scientific) using an EASY-Spray ion source. All samples were stored in their lyophilized form and were resuspended by the autosampler prior to injection as 1 μg sample material was loaded onto an Acclaim PepMap 100 trap column (75 μm × 2 cm, C18, 3 μm, 100 Å), washed for 5 min at 0.250 μl/min with Solvent A (95% $H_2O$, 5% DMSO, 0.1% FA) and thereafter separated using a PepMap 800 C18 column (15 cm × 75 μm, 3 μm). The gradient went from Solvent A to Solvent B (90% ACN, 5% $H_2O$, 5% DMSO, 0.1% FA) at a constant flow of 0.250 μl/min, up to 43% Solvent B in 40 min, followed by an increase up to 55% in 10 min followed by a steep increase to 100% B in 2 min. Online LC-MS was performed using a Q-Exactive HF mass spectrometer (Thermo Scientific).

**Spectral library generation for targeted proteomics**. An equimolar pool of QPrESTs (Supplementary Table 6) was digested by trypsin according to the in-solution protocol described above. Only non-secreted protein targets were included from a set of QPrEST sequences previously generated based on being frequently identified in shotgun proteomics experiments performed on cell lines. The recombinant protein sequences had to generate at least one proteotypic peptide upon trypsin digestion. For initial screening, 50 fmol of each QPrEST was loaded onto column and a Top5 MS method was performed with master scans at 60,000 resolution (mass range 300–1600 m/z, AGC 3e6), followed by five consecutive MS/MS scans at 30,000 resolution (AGC 1e5, underfill ratio 0.1%) with normalized collision energy set to 25. Raw mass spectrometry files were processed using MaxQuant, using the search engine Andromeda and searched against a FASTA file containing QPrEST sequences spiked into the E. coli proteome (BL21, Uniprot-ID: #UP000002032), which was used for recombinant protein production Identified peptides were further processed by only allowing proteotypic peptides mapping to one single human gene (defined by Swiss-Prot v72), thereby excluding peptides with potential miscleavages and peptides including any methionine.

**Data-independent MS acquisition targeted proteomics**. Full MS master scans at 60,000 resolution (mass range 300–1600 m/z, AGC 1e6) were followed by 20 data-independent acquisitions MS/MS at 60,000 resolution (AGC 1e6) defined by a scheduled Parallel Reaction Monitoring (PRM) method (Supplementary Data 10). Precursors were isolated in a scheduled sequence with a 1.2 m/z isolation window and the maximum injection time was set to 105 ms for both MS and MS/MS resulting in a duty cycle of 2.7 s. The isolation list was split into two consecutive LC-runs, targeting 120 paired light and heavy peptides per injection.

**MS data evaluation and protein quantification**. Raw MS-files from the data-independent method were processed using Skyline Proteomics Environment[24]. The ratio between endogenous and heavy peptide standard was calculated from the summed area intensity over retention time for each peptide fragment separately (Supplementary Data 11). Median peptide ratios between replicates were used to calculate the amount of peptide present in the cell lysate. When more than one peptide per protein was available (12 proteins = 1 peptide, 14 proteins = 2 peptides, 2 proteins = 3 peptides, 1 protein = 4 peptides), the median peptide value was used to determine the relative protein amount (Supplementary Data 4).

**Western blot analysis I**. Lysates representing the same number of cells, approximately 15 μg of protein, from the cell lines A431, A549, HEK 293, U-2 OS, Hep G2, HeLa, RT4, and SH-SY5Y subjected for PRM analysis were, together with Precision Plus protein standard (Bio-Rad Laboratories, Hercules, CA, USA), loaded on precast 4–20% Criterion TGX Precast Gels (Bio-Rad Laboratories) and run under reducing conditions. Protein transfer to PVDF membranes was performed using Trans-Blot Turbo Transfer system (Bio-Rad Laboratories) and the mem-branes were blocked for 1 h at room temperature (RT) (5% dry milk, 0.5% Tween 20, TBS; 1 mM Tris-HCl, 0.15 M NaCl). The membranes were incubated with primary antibody (0.2 μg/ml) for 1 h at RT, followed by incubation with horse-radish peroxidase (HRP)-conjugated antibody (swine anti-rabbit 1:4000, P0399, Dako) for 45 min at RT. Detection and quantification of the signal from the chemiluminescence substrate (Immobilon Western Chemiluminescence HRP Substrate, Merck Millipore) was made using a ChemiDoc XRS + CCD camera and the software Image Lab (Bio-Rad Laboratories). After detection, the membranes were stripped using a protocol for mild stripping (Abcam, Cambridge, UK) and blocked for 1 h. The membranes were incubated with anti-alpha tubulin antibody (0.05 μg/ml, ab4074, Abcam) for 1 h at RT followed by incubation with HRP-conjugated antibody and signal detection as above (Supplementary Data 5).

**Western blot analysis II**. In order to cover the standard HPA cell lines (RT4 and U-251), four cell lines were replaced from Western blot analysis I. Approximately 15 μg of protein from the cell lines RT4, U-251, A549, Hep G2, U-2 OS, BJ, CACO-2, and SK-MEL-30 subjected for TMT analysis were prepared by mixing with reducing buffer (1.5 mg/ml lysate, Laemmli Sample Buffer (Bio-Rad Laboratories), 50 mM DTT) and heated to 95 °C for 5 min. The lysates were then loaded on a precast 4–20% Criterion TGX Stain-Free Gel (Bio-Rad Laboratories) together with PageRuler Plus Prestained Protein Ladder (Thermo Scientific), and run in Tris/Glycine/SDS Electrophoresis Buffer (Bio-Rad Laboratories). The gel was imaged using Stain-Free imaging with a ChemiDoc MP CCD camera and the Image Lab software (Bio-Rad Laboratories). To ensure that the loaded amount of protein was equal for all cell lines, the loading volume of each cell line was adjusted according to the results from the Stain-Free gel imaging and new gels were run for membrane production. Protein transfer to PVDF membranes was performed using the Trans-Blot Turbo Transfer system as described by the manufacturer (Bio-Rad Labora-tories) and the membranes were blocked for 1 h at RT (5% dry milk, 0.1% Tween 20, TBS; 10 mM Tris, 0.15 M NaCl). The membranes were incubated with primary antibody (0.4 μg/ml) for 1 h at RT, followed by washing in TBS (1 × 5 min) and incubation with HRP-conjugated antibody (swine anti-rabbit 1:3000, P0399, Dako) for 45 min at RT. After additional wash with TBS (3 × 5 min) detection and sub-sequent quantification of the signal (Immobilon Western Chemiluminescence HRP Substrate, Merck Millipore) was performed using a ChemiDoc MP CCD camera and the Image Lab software (Bio-Rad Laboratories). To verify equal loading, the membranes were thereafter incubated with an anti-PSMA1 antibody (Supple-mentary Fig. 9), which served as an internal standard (0.3 μg/ml, HPA037646) and HRP-conjugated antibody (swine anti-rabbit 1:3000, P0399, Dako) mixture for 1 h at RT followed by signal detection as above (Supplementary Data 5).

**Cell line cultivation for TMT quantification and capture MS**. A549 and BJ cells were grown in Dulbecco's modified Eagle medium (DMEM; Invitrogen, Carlsbad, CA) supplemented with 10% FBS and 1% streptomycin. SK-MEL-24 cells were grown in EMEM supplemented with 15% FBS and 1% streptomycin. CACO-2, Hep G2, SK-MEL-28 and U-251 were grown in EMEM supplemented with 10% FBS and 1% streptomycin. RT4 and U-2 OS were grown in McCoy supplemented with 10% FBS and 1% streptomycin.

**Cell lysis for TMT quantification and capture MS**. Cells were lysed in lysis buffer (1% SDC in 1× PBS supplemented with 1 mM sodium orthovanadate, 1 tablet of Complete mini EDTA-free mixture (Roche Applied Science) and one tablet of PhosSTOP phosphatase inhibitor mixture (Roche Applied Science) per 10 ml of

lysis buffer) by heating at 95 °C for 10 min. Cells were further lysed by 10 rapid passages through a 23-gauge hypodermic syringe needle and by sonication on ice. After centrifugation (20,000 × g, 30 min at 4 °C), the protein concentration was determined by BCA (Pierce).

**Sample preparation for capture MS.** Protein lysates from the two cell lines (RT4 and U-251) were prepared in Laemmli Sample Buffer (BioRad Laboratories, 50 mM DTT) to a concentration of 1.5 mg protein per ml lysate and heated to 95 °C for 5 min. Approximately 45 μg of protein was loaded, in triplicates, on a precast 4–20% Criterion TGX StainFree Gel (BioRad Laboratories), together with PageRuler Plus Prestained Protein Ladder (Thermo Scientific), and run according to the manufacturer's recommendation. The gel was rinsed in water (3 × 5 min), stained using GelCode Blue Stain Reagent (Thermo Scientific) for 1 h and destained through washing in water (2 × 1 h). Lanes were excised in bands according to Supplementary Fig. 10. The in-gel digestion was performed as previously described[25] with some changes to the original protocol as outlined below. Briefly, gel pieces were shrunk in 100% ACN and reduced by 10 mM DTT for 30 min at 56 °C. Alkylation was performed by addition of 55 mM 2-choloroacetamide and incubated in dark for 20 min. Trypsin digestion was performed over night at 37 °C after addition of trypsin (13 ng/μl in 50 mM ammonium bicarbonate, 10% ACN (vol/vol)). Peptides were extracted by addition of 100 μl of extraction buffer (1:2 (vol/vol) 5% formic acid/ACN) and dried down by vacuum centrifugation prior to LC-MS/MS analysis.

**LC-MS/MS analysis of gel pieces.** Each sample was analyzed on a Bruker Impact II (Bruker Daltonics, Germany) connected to a Dionex UHPLC system (Thermo Fisher Scientific). The UHPLC was equipped with a trap column (Acclaim PepMap 100, 75 μm × 2 cm, nanoviper, $C_{18}$, 3 μm, 100 Å; Thermo Fisher Scientific) and an analytical column (PepMap RSLC $C_{18}$, 2 μm, 100 Å, 75 μm × 50 cm; Thermo Fisher Scientific). Mobile-phase buffers for nLC separation consisted of 5% ACN/0.1% FA in water (solvent A) and 95% ACN/0.1% FA (solvent B). The peptides were eluted during a 2 h gradient and directly sprayed into the mass spectrometer. The flow rate was set at 400 nl/min, and the LC gradient was as follows: 4% solvent B within 5 min, 4–32% solvent B within 90 min, 32–95% solvent B within 1 min, 100% B for 10 min and down to 2% solvent B within 1 min following equilibration at 2% solvent B for 13 min. Nano spray was achieved with an applied voltage of 1.6 kV using a Captive Spray source (Bruker Daltonics, Germany) connected to a NanoBooster filled with 100% ACN. The mass spectrometer was programmed in a data-dependent acquisition mode (cycle time set to 3 s) and was configured to perform a survey scan from 150 to 2200 $m/z$ (1 Hz). MS2 scans were acquired in dynamic mode (2500 cts = 8 Hz to 25,000 cts = 32 Hz) of ions with charge state between 2 and 5 with a smart exclusion (5×) set to 30 s.

**Data analysis of capture MS.** Intensities for label-free MS quantification was performed by analyzing the raw data by MaxQuant (version 1.5.7.0)[26]. Andromeda[27] was used to search the MS/MS data against the Ensembl *Homo sapiens* database (version 88.38, using all protein-coding transcripts from the primary assembly) complemented with a list of common contaminants and concatenated with the reversed version of all sequences. Trypsin/P was chosen as cleavage specificity allowing two missed cleavages. Carbamidomethylation (C) was set as a fixed modification, while oxidation (M) was used as variable modification. Data filtering was carried out using the following parameters: peptide and protein FDRs were set to 1%, minimum peptide length was set to 7. The reverse and common contaminant hits were removed from the MaxQuant output.

**Protein digestion prior to TMT labeling.** Proteins were reduced with 2 mM DTT at RT for 1 h, alkylated with 4 mM 2-chloroacetamide at RT for 30 min in the. A first enzymatic digestion step was performed Lys-C at 37 °C O.N. (1:50 w/w). Each sample was diluted ten times and digested with trypsin O.N. (1:50 w/w). The resulting peptides of each cell line were chemically by TMT10plex according to the manufacturer's instructions: Reference Pool (126); SK-MEL-24 (127 N); SK-MEL-28 (127 C); U-2 OS (128 N); A549 (128 C); U-251 (129 N); BJ (129 C); RT4 (130 N); Hep G2 (130 C); CACO-2 (131). Samples were mixed 1:1 and desalted using Sep-Pak Vac $C_{18}$ cartridge (1 cc/200 mg, Waters), and the eluted peptides were dried down and stored at −20 °C. The same procedure was repeated for the other two biological replicates.

**Basic reverse phase fractionation.** Two hundred μg of peptide mixture from each set were fractionated using a Waters XBridge BEH300 C18 3.5 μm 2.1 × 250 mm column on an Agilent 1200 series operating at 200 μl/min. Buffer A consisted of 20 mM $NH_3$, while buffer B of 80% ACN/20 mM $NH_3$. The fractionation gradient was: 3–88% B in 63 min; 88% B for 15 min; and ramped to 100% B in 2.5 min; 100% B for 13.5 min. Fractions were collected into polypropylene V-96 well microtiter plates (Microplate, 96 well PP, V-Bottom; Grainer BIO-ONE). At 97 min, fraction collection was halted, and the gradient was held at 3% B for 20 min. The total number of concatenated fractions was set to 12. Each plate was dried at RT using a Speed Vac (SPD 111 V, Thermo). Plates were stored at −20 °C till LC-MS/MS analysis.

**LC-MS/MS analysis of TMT-labeled samples.** Each sample was analyzed on a HF Q-Exactive Orbitrap (Thermo Fisher, Germany) connected to a Dionex UHPLC system (Thermo Fisher Scientific). The UHPLC was equipped with a trap column (Acclaim PepMap 100, 75 μm × 2 cm, nanoviper, $C_{18}$, 3 μm, 100 Å; Thermo Fisher Scientific) and an analytical column (PepMap RSLC $C_{18}$, 2 μm, 100 Å, 75 μm × 50 cm; Thermo Fisher Scientific). Mobile-phase buffers for nLC separation consisted of 0.1% FA in water (solvent A) and 80% ACN/0.1% FA (solvent B). The peptides were eluted during a 2 h gradient and directly sprayed into the mass spectrometer. The flow rate was set at 250 nl/min, and the LC gradient was as follows: 3–6% solvent B within 3 min, 6–35% solvent B within 117 min, 35–47% solvent B within 5 min, 47–100% solvent B within 5 min and 100% B for 8 min and 1% solvent B for 5 min. Nano spray was achieved with an applied voltage of 1.8 kV. The mass spectrometer was programmed in a data-dependent acquisition mode (top 10 most intense peaks) and was configured to perform a Fourier transform survey scan from 370 to 1600 $m/z$ (resolution 60,000), AGC target 3e6, maximum injection time 250 ms. MS2 scans were acquired on the 10 most-abundant MS1 ions of charge state 2–7 using a Quadrupole isolation window of 1 $m/z$ for HCD fragmentation. The collision energy was set at 34%; resolution = 30,000; AGC target 2 e[5], maximum injection time 200 ms; dynamic exclusion 15 s.

**Data analysis.** TMT10plex quantification was performed by analyzing the raw data by MaxQuant (version 1.5.3.30)[26]. Andromeda[27] was used to search the MS/MS data against the UniProt *Homo sapiens* database (containing canonical and isoforms_42144 entries downloaded on 21 March 2016) complemented with a list of common contaminants and concatenated with the reversed version of all sequences. Trypsin/P was chosen as cleavage specificity allowing two missed cleavages. Carbamidomethylation (C) was set as a fixed modification, while oxidation (M) was used as variable modification. Data filtering was carried out using the following parameters: peptide and protein FDRs were set to 1%, minimum peptide length was set to 7. The reverse and common contaminant hits were removed from MaxQuant output. Samples were normalized to the median intensity for each TMT channel respectively after removing CON and REV protein sequences (Supplementary Data 9). Protein values used for orthogonal validation were calculated across triplicate LC-MS/MS measurements and reported as the median protein intensity (Supplementary Data 3).

**Transcriptomics analysis.** Procedures for extraction of RNA, library preparation, and sequencing has been described elsewhere[29]. Reads were mapped to the human reference genome assembly GRCh38 and quantified by Kallisto version 0.42.4[30]. Normalized expression levels (TPM values) on gene level were obtained by summing the estimated values from the constituent transcripts of each gene, respectively.

## Data availability

The TMT10plex data have been deposited to the ProteomeXchange Consortium (http://proteomecentral.proteomexchange.org) via the PRIDE partner repository[28] with the data set identifier PXD005914. The Capture MS LC-MS/MS data can be accessed using the PXD010178 identifier. Chromatograms used for the targeted proteomics quantification are made available through Panorama (https://panoramaweb.org/western_blot.url) and the corresponding raw data are available through Chorus (chorusproject.org), experimental ID 1213. All cell line data from transcriptomics analysis are available at http://www.ncbi.nlm.nih.gov/bioproject/PRJNA183192. All cell line data are available at http://www.ncbi.nlm.nih.gov/bioproject/PRJNA183192.

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

## Acknowledgements

We acknowledge the entire staff of the Human Protein Atlas program and the Science for Life Laboratory for valuable contributions. Funding was provided by the Knut and Alice Wallenberg Foundation and Erling-Persson Foundation to MU. Correspondence and requests for materials should be addressed to MU.

## Author contributions

F.E., A.H., K.L., G.M. and A.A. performed experimental work. F.E., A.H., G.M., B.F., L.F. and Å.S. analyzed the data. Å.S., H.T., S.H., L.F., C.A.K.S., K.v.F., P.O., and M.U. generated the antibodies as part of the Human Protein Atlas program. S.H., K.vF., C.L., and C.A.K.S. provided intellectual input. F.E., A.H., G.M., B.F., and M.U. wrote the manuscript.

## Additional information

**Competing interests:** M.U. and S.H. are co-founders of Atlas Antibodies and M.U., P.O. and K.v.F. are co-founders of Antibodypedia. B.F. acknowledges formal links to Atlas Antibodies. The remaining authors declare no competing interests.

