## [Peer Review File · Nature Communications]

Reviewer Comments:

Reviewer #1 (Remarks to the Author):

The authors have addressed all my outstanding concerns.

Reviewer #2 (Remarks to the Author):

Overall we think that the author improved many of the formal criticism we and other reviewers raised. However the major criticism remains since we don't see how their work significantly addressed the reproducibility crisis in preclinical biomedical research using unspecific antibodies for detection of human proteins. The consequences of relaxed validation criteria for the reproducibility of preclinical studies can be quite fatal. For example in a recent study (Kosmidou C. et al., 2018, Scientific Reports, 7:461) to find out potential causes of contradicting published results regarding the protein NLRP3 it has been shown that out of nine antibody reagents against NLRP3 used in a multitude of published studies only one turned out to be specific. The eight unspecific ones included the antibody HPA012878 assigned as validated by one of the methods in the presented manuscript (indicated in supplementary table S8). This example on the antibody specificity issue could only be resolved by the corresponding knock out control experiment. It is of course a single example and we don't know at hand how widespread this problem currently is in the biomedical literature but it clearly underlines the need for rigorous genetic validation when using antibodies for biomedical research.

Despite the access to high throughput knock down technologies the authors unfortunately were not considering this method as their major pillar in their validation but included a number of quite "coarse" methods of unknown FDR. We therefore think the claimed validation of 6000 antibodies is overstated as the presented data rather represent a screen for good antibody candidates which which should not be used without rigorous validation by the individual researchers. If despite these limitations the editors consider the manuscript for publication in Nature methods I think it is essential that the authors clearly state in the final section of their discussion that the data provided represent an initial orientation for choosing antibody reagents and that it remains the responsibility of the individual researchers to perform rigorous specificity testing by genetic knock down experiments on the antibodies validated in this manuscript to ensure antibody specificity.

specific responses to the rebuttal letter

Reviewer 2

No information is provided how many of the detected proteins show significantly altered protein expression across cell lines chosen for TMT and PRM analysis respectively.

Answer: This has been summarized in a new Table S6 (including max, min, total FC, CV across cell lines and number of cell-lines with missing data).

Response: Table S6 does not provide information on the number replicates being performed

Why has a different panel been chosen for TMT and PRM analysis?

Answer: Cell-line panels were initially chosen based on availability for the PRM evaluation (i.e. based on what cell-lines were grown by the human protein atlas at time of analysis). Selection were initially based on transcriptomics profiling of 56 cell-lines (RNA-seq) (Uhlén et al, 2015, Science). No direct comparison was made between TMT and PRM as each dataset was compared to the WB-results. We think that this rationale, using two methods, show a flexible way forward for orthogonal protein quantification. Most importantly, the rationale for changing cell-lines between the PRM and TMT dataset was to include both RT4 and U-251 as they were missing in the original PRM-dataset. This to enable us to compare all Western Blots performed within HPA to the TMT results. Noteworthy, four cell-lines remain the same across both the TMT and PRM dataset. We have added a section in the materials and methods clarifying this.

How many of the quantified proteins display significant abundance changes across the chosen cell line panel? This certainly has a direct impact on the fraction of the antibodies that can be evaluated by this approach. Also no explanation is presented on how the arbitrary 0.5 Pearson correlation cutoffs were chosen to validate antibody specificity. What is the variation in Western blot signals across the cell lines measured in biological triplicates and has this been considered in the chosen cutoffs? No p-value statistic is provided for the orthogonal validation methods.

Answer: We have avoided statistical term as much as possible. The wording “arbitrary Pearson correlation cutoff” has been chosen simply since it is arbitrary. A correlation coefficient of 0.5 is often considered moderate and correlations below 0.5 are thus considered low. We have not explored the optimal correlation coefficient for different sized panels as this is outside the scope of this study.

Migration capture MS

Most proteins migrate between 30-60 kDa and therefore detection of a western blot bands in this size range may not be a stringent criteria for antibody specificity in this crowded size range. Also the author present no information on how well actually the migration capture MS data correlate between the two cell lines measured or between their data and the previously published data to estimate the error associated with the method. In addition it is unclear how exactly the authors correlate the size measured in the MS based “virtual Western” with the size detected by the antibody?

Answer: This has been clarified in the revised manuscript. This was done using the ladder visualized in Figure S10. The experiment was performed in triplicate, including cutout of gel pieces. The replicate results can be seen on the validation page for each antibody respectively.

What is the delta size that they still accept as good correlation between migration capture MS and Western blotting?

Answer: We are not correlating the intensity with the bands and this has been clarified in the manuscript. An antibody is validated if the peak intensity from Capture MS reveals the correct position thereby providing higher specificity than the conventional method by comparing the band to the theoretical size. We have included a new Figure 2c highlighting that 2,054 antibodies validated by Capture MS also are validated using another validation pillar.

Response: it is still not clear how the authors claim the correct position in Capture MS: is this based on the correlation with Western blot signal or the predicted size. How do the authors deal with cases when multiple bands are observed by the antibody?

Size determination by Western blotting may be associated with a significant error which adds to the noise in the correlation and thus reduces that rigor of specificity. In the absence of a good model that estimates the FDR for a protein being detected by chance in the gel slices, migration capture MS validation remains a very coarse method in estimating antibody specificity and I would not recommend using antibodies validated by this method alone.

Answer: We agree that it is a coarse method and definitely the weakest of the pillars. We have clarified this point in the revised manuscript.

Genetic strategies

One of the most reliable ways to test whether an antibody is specific is to compare lysates from cells that either express or do not express the corresponding antigen. To probe antibody specificity the authors used siRNA knock downs of their targets proteins. For validation they accept an arbitrary reduction of 25% in signal by at least one of the knock down reagents as sufficient for validation. I think this is not stringent enough for validation given limited quantitative accuracy in Western blotting. In most published siRNA Western blotting experiments a reported signal reduction of 80% or higher is quite common. CRISPR/CAS9 mediated knock out cell lines could serve as alternatives tool but has not been used. Unfortunately, only a small fraction of their antibodies have been validated by the most informative and stringent validation method.

Answer: We agree that this is a good way to validate antibody, but often only a single cell line is used for the validation which is a limitation. The method is also quite labor intensive (in particular if gene editing, such as CRISPR-Cas9, is used) and therefore not easy to scale. In fact, the knock-down experiments reported in the manuscript involved a considerable effort and is as far as we know the largest genetic validation of antibodies reported so far.

Response: siRNA knockdowns can be done at high throughput. There is no need to perform the same experiment in another cell line in order to demonstrate specificity if the protein of interest is expressed in a given cell line. No answer is provided why a 25% reduction suffices for the validation in light of the known limitations in quantitative accuracy by Western blotting. This sounds to me quite arbitrary and should be justified.

Integration of the validation results and conclusion

Sentences such as “the choice of suitable ambition level needs to be discussed by the various stakeholders” at the end of the discussion do not help either when the community needs to address the antibody specificity issue. The ambition is clear in my view: the term validation should only be exclusively used with the most specific validation tool (siRNA knock down) which will cut down significantly the list of 1.5 Mio non validated research antibodies. If the broader biomedical research community aims to overcome the reproducibility crisis caused by lack of specific Western blotting antibodies then the presented study in its current form does not represent a significant step forward in my view.

Answer: We agree that the genetic methods are excellent for showing specificity, but it isn't applicable in all instances and has low throughput.

Response: it has been demonstrated that siRNA knock down experiments can be applied genome wide and work at high throughput.

Reviewer #3 (Remarks to the Author):

As I indicated in my original review, this is an impressive amount of data and I fully agree with the authors' attempts to improve the validation of antibodies and to provide the data they generate to the scientific community. This revision more clearly describes the experiments and the pros and cons of each pillar. In my original review, I indicated that the novelty required to publish in Nature Methods would be the systematic comparison of the effectiveness of each pillar and I still find this lacking. The issues that remain are listed below:

Major points:

1) As I mentioned in point 1 in my prior review – the dataset was not chosen to consist of a set of known true positives and negatives to perform a systematic comparison. The set of 6000 antibodies has gone through preliminary validation as part of the HPA. This may affect the success rate of the pillars being evaluated if the initial validation is more similar to one of the pillars. Furthermore, there is not one single antibody that has been evaluated by all 5 pillars (Table S8). Figure 4b compares across pillars but only for those that have shown enhancement in 3 or more pillars. It would have been better to look at those that have been tested in 3 or more pillars (1146 antibodies) to compare between them.

2) The authors state that the antibodies are validated if they pass one of the pillars (line 316). From table S8 – there are 4384 that have been validated by at least one pillar. Of these, 1560 have had at least one other test: 1165 with one 'uncertain', 359 with two 'uncertains' and 36 with 3 other 'uncertains'. What is the justification for suggesting an antibody has enhanced its validation when it has been unsuccessful in the same number or more pillars than it has been successful? The high success rate of Capture MS compared to the more specific Genetic Strategy suggests that there is an issue with false positives. The authors' also have not clearly indicated how the siRNA used to assess the antibodies were validated. If they are using unvalidated reagents to validate antibodies, then it is probable that their genetic strategy has false negatives (as they eluded to in lines 278-280).

3) The authors prefer not to rate anything as 'failed verification' so the reader cannot easily discern if the 'uncertain' is due to limitations of the assay or the antibody. At a minimum, the authors should differentiate between those that are uncertain because of a limitation of the assay (low expression variability for orthogonal) and those that have met the criteria for the assay and still do not pass ('failed verification').

Other points:

4) Western blots do not have replicates so there is no assessment of the variability

5) Figures 2D and S4 are still confusing.

a) Figure 2D – For the data - the blue colors label 'experiment types' but your red colour labels a 'result' leaving it unclear if the blue marks are 'all tested' or just those classified as 'enhanced' and if the red marks mean failure in 'proteomics' or 'transcriptomics'.

b) Figure S4 – remove the grey cutoff as it applies to the fold change cutoff for transcriptomics. As shown by PRM and TMT – the dynamic range can be quite different in fold change cutoffs – so it does not make sense to overlay a cutoff from one experiment to another and does not make sense here as no fold-change filter was applied. Then as for Figure 2D – clarify your data colors as it is not clear why the points below 0.5 Pearson are not red in the TMT graph. Also, it would be good to label those with antibody name similar to the PRM.

6) Line 168 – states that only 6 of the antibodies could not be validated because of low expression variability – this implies that those above the 0.5 Pearson were validated. If there is too much statistical noise in this area – then why are these considered validated instead of uncertain?

7) Figure S5 – these are two clear cut examples. The authors should include examples where there is less than 5-fold difference. Error bars also appear to be missing from the proteomics.

8) Supplementary tables – I would suggest adding the gene name where appropriate and not just the antibody ID. In Table S9 – it implies that the antibody is being measured and not the expected protein target.